# Red Algae Alters Expression of Inflammatory Pathways in an Osteoarthritis *In Vitro* Co-Culture

**DOI:** 10.3390/ph18030315

**Published:** 2025-02-24

**Authors:** Shane M. Heffernan, Mark Waldron, Kirsty Meldrum, Stephen J. Evans, Gillian E. Conway

**Affiliations:** 1Applied Sports Science Technology and Medicine Research Centre (A-STEM), Faculty of Science and Engineering, Swansea University, Swansea SA1 8EN, UK; mark.waldron@swansea.ac.uk; 2In Vitro Toxicology Group, Faculty of Medicine, Health and Life Sciences, Swansea University, Swansea SA2 8PP, UK; kirsty.meldrum@swansea.ac.uk (K.M.); s.j.evans@swansea.ac.uk (S.J.E.); gillian.conway@swansea.ac.uk (G.E.C.)

**Keywords:** Red Algae, gene expression, inflammation, osteoarthritis, *in vitro*

## Abstract

**Background/Objectives:** Osteoarthritis (OA) is one of the most prevalent chronic conditions and significantly contributes to local and global disease burden. Common pharmaceuticals that are used to treat OA cause significant side effects, thus non-pharmaceutical bioactive alternatives have been developed that can impact OA symptoms without severe side-effects. One such alternative is the Red Algae *Lithothamnion* species (Litho). However, there is little mechanistic knowledge of its potential to effect OA gene expression, and a human in vitro model using commercially available cell lines to test its effectiveness has yet to be developed. **Methods:** Human osteoblast (hFOB 1.19. CRL-11372) and chondrocyte (C28/I2) cell lines were co-cultured indirectly using transwells. IL1-β was used to induce an inflammatory state and gene expression profiles following treatment were the primary outcome. **Conclusions:** Results indicated that the model was physiologically relevant, remained viable over at least seven days, untreated or following induction of an inflammatory state while maintaining hFOB 1.19. and C28/I2 cell phenotypic characteristics. Following treatment, Litho reduced the expression of inflammatory and pain associated genes, most notably *IL-1β*, *IL-6*, *PTGS2* (*COX-2*) and *C1qTNF2* (*CTRP2*). Confirmatory analysis with droplet digital PCR (ddPCR) revealed that Il-1β induced a significant reduction in *C1qTNF2* at 7 days which was ameliorated with Litho treatment. These data present a novel and replicable co-culture model of inflammatory OA that can be used to investigate bioactive nutraceuticals. For the first time, this model demonstrated a reduction in *C1qTNF2* expression that was mitigated by Red Algae *Lithothamnion* species.

## 1. Introduction

Osteoarthritis (OA) is one of the most prevalent chronic conditions globally and contributes significantly to disease burden [1,2]. It is estimated that OA effects 595 million individuals worldwide [3,4,5] and ~13% of those over 50 years [6]. OA is a degenerative disease caused by the failure of normal biological processes to repair damaged tissue leading to abnormalities in synovial joints [7], such as subchondral osteophytes [8], local inflammation or synovitis [9,10,11,12], bone marrow lesions [13] and systemic low-grade inflimation [14]. It is well established that inflammatory processes are key drivers of OA, often exacerbated by immune system activation particularly in older populations, but not exclusively [15,16].

There is no known cure for OA [17,18,19,20,21], however drug therapy, physical therapy and surgery are common treatments [22]. In the early stages, pain relief and inflammatory reduction can be achieved through appropriate exercise and weight loss strategies. However, adherence is often challenging [23] and these methods are commonly accompanied by oral non-steroidal anti-inflammatory drugs (NSAIDs) and other analgesics [24,25]. Current global data shows that four in every ten OA patients seeking healthcare are prescribed some form of NSAID [26], which is not surprising as it is currently a “strongly recommended” strategy for OA clinical management [27,28]. Further, OA patients frequently use Paracetamol or acetaminophen, N-acetyl-p-aminophenol [29] (34%; in isolation or in combination with NSAIDs) and opioids (8–26%), the volume of which has been described as “alarmingly high” and shown to be inappropriately dispensed [30].

All these common pharmaceuticals can cause significant side effects. While some NSAIDs are effective at improving pain symptoms and physical function, their regular use can result in gastrointestinal complications, renal disturbances and severe cardiovascular events [31]. Paracetamol may be ineffective for treating some types of OA pain [32,33], but it has similar side effects as ibuprofen [34], particularly when consumed at higher doses [35], and its overuse can cause liver injury, hepatotoxicity, mitochondrial toxicity [36,37]. The negative health effects of opioid use are well documented, but they are frequently prescribed for OA and usage is expected to triple in the coming years [30,38]. Critically, a number of systematic reviews and meta-analyses have reported that the tolerability of opioids is low, efficacy for pain relief in OA is not clinically relevant and the potential risks of harm are high [39,40,41]. Therefore, non-pharmaceutical bioactive alternatives, that can reduce pain but without the severe side-effects, could be effective early interventions and can be used in isolation or alongside pharmaceuticals [24,42,43,44] to improve OA symptoms [45,46,47,48]. However, to date there is no readily accessible physiologically relevant *in vitro* model to test their potential molecular impact (morphology, inflammation etc.) or safety [49,50]. There is a need to develop such a model using human cell lines that addresses some current methodological limitations and to reduce the burden of animal testing [50], particularly in OA [49].

One non-pharmaceutical bioactive alternatives, Red Algae *Lithothamnion* species have been reported, individually and in combination with other bioactives, to improve OA symptoms and functional performance *in vivo* [51,52,53,54]. Two double-blinded randomised trials utilising *Lithothamnion* species [53,54] showed improved moderate-to-severe knee OA pain, symptoms (stiffness) and functional performance (6 m walking distance) compared to Glucosamine [54], and when NSAIDs were reduced by ~50% *Lithothamnion* treatment improved physical performance [53]. Early mechanistic work suggested this may be a result of inhibited NFκB pathway, reduce tumour necrosis factor alpha (TNF-α), interleukin 1 beta (IL-1β) and COX-2, along with reduced serum TNF-α [52,54,55,56]. However, it remains unclear to what extent *Lithothamnion* can mediate the inflammatory response of OA in a human physiologically relevant *in vitro* model and what pathways are of particular importance.

Therefore, the purpose of the present study was to develop an accessible, cost effective, novel *in vitro* human co-culture model, intended for nutraceutical investigation and to assess the effect of *Lithothamnion* species on the inflammatory response to an OA environment.

## 2. Results

### 2.1. Development of Co-Culture Model

As a monoculture, hFOB 1.19. cells were characterised by Von Kossa staining, Alizarin Red (ARZ) and Alkaline phosphatase (ALP) activity. There was an increase in mineralisation and ALP activity, which remained consistent across time (Figure 1A–C). C28/I2 cells were characterised by relative gene expression of *COL1A1*, which increased across time and the presence of aggrecan staining was shown via confocal microscopy (Figure 1D,E).

For co-culture, both hFOB 1.19. and C28/I2 cells demonstrated typical growth patterns for each cell type and morphology was consistent with healthy cells. This was also confirmed by presto blue viability data across 7 days (Figure 2A). Light and confocal Microscopy confirmed that hFOB 1.19. growth and C28/I2 cells maintained their location on the transwell inserts and typical growth characteristics (Figure 2B). Cell viability remained high (>95%) for co-culture (C28/I2 and hFOB 1.19. cells) and cell size consistent (~16.5 μm) across time. To induce inflammation, thus simulating an osteoarthritic-like scenario (OA model) [57], IL-1β was added to the co-culture at 10 ng/mL, viability (>93%) and cell size remained (~16.5 μm) consistent with no significant drop in viability over 7 days (Figure 2A). The presence of inflammation was confirmed by IL-6 and IL-8 ELISAs, markers known to be present and often used in OA experimental designs [14,57]. There was a significant increase in IL-6 between day 1 and day 7. In the presence of IL-1β, IL-8 increased by 3.6 and 3.3 fold for day 1 and day 7 respectively and IL-6 increased by 17.7 and 2.3 fold for day 1 and day 7 respectively, compare to untreated media condition (Figure 2C,D).

### 2.2. Effect of Treatment on Co-Culture Phenotype

Cell viability assay demonstrated that Litho had no impact on cell viability until beyond 2 mg/mL for hFOB 1.19 cells and up to 1 mg/mL for C28/I2 cells across the experimental period when grown as monocultures (Figure 3A,B). In co-cultures, there was no difference in cell viability when treated with IL-1β CM, or IL-1β combined with Litho. There was a small drop in viability for cells treated with Litho 1.0 mg/mL (87%), however this recovered over the course of the 7 days (97%) (Figure 3C,D). Litho at all doses significantly reduces IL-6 (*p* < 0.006) compared to untreated controls at day 7, but not with added IL-1β CM, although there was a tendency for the 0.25 mg/mL dose to be lower than IL-1β media (*p* = 0.07; Figure 3E–H). There was no effect of treatment dose on IL-8, with or without IL-1β CM.

### 2.3. Impact of Treatment Co Culture Gene Expression

For co-culture with IL-1β CM, *IL-1β* showed the greatest change in normalised expression, with an 18.0 fold increase, reducing to 3.84 fold following 7 days of repeated dosing. *IL-6* expression increased 11.86 fold at day 1 and reduced to 2.50 fold after 7 days. *ITGB2* increased by 6.54 fold, reducing to −1.02 after 7 days. *PTGS2* was upregulated by 8.37 fold compared to untreated control (IL-1β CM), reducing to 2.0 fold after 7 days. Interestingly, *GPR22* was downregulated, albeit not biologically relevant at day 1 (−1.17 fold), but substantially upregulated at day 7 (14.20 fold; Figure 4A). Following treatment with experimental material, Litho reduced the expression of *IL-1β* by ~16 fold, *IL-6* by ~10 fold, *PTGS2* by ~7 fold and *ITGB2* went from being upregulated (6.54 fold) to downregulated (~−6.0 fold) for both doses at day 1. Following 7 days of repeated dosing, *GPR22* went from being upregulated (14.20 fold) to downregulated (−3.94 fold) in both doses (Figure 4B).

Seven genes were selected for confirmatory analysis by ddPCR following pathways of interest that were derived from qPCR array data (pain and inflammation). Data from three genes (*CCL2*, *TGF*β*3* and *PTGDS*) were below detectible limits, thus were not statistically analysed or presented. Of the remaining genes (Figure 4C), there was no statistical difference observed for *COL1A1* or *TNFSF10*, however at day 7, the addition of IL-1β appeared to reduce the expression of *COL1A1* (*p* = 0.15), which was unaffected by Litho (*p* < 0.50). The introduction of IL-1β CM increased *IL-6* and *PTGS2* expression on day 1 and 7 (*p* = 0.003 and *p* = 0.034; *p* = 0.032 and *p* = 0.022, respectively) but again were unaffected by Litho treatment (*p* < 0.39). *PTGS2* expression was significantly affected by time in each experimental condition (greater expression on day 7; *p* > 0.006). IL-1β induced a significant reduction in *C1QTNF2* compared to unstimulated control at day 7 (*p* = 0.048), however this reduction was returned to untreated levels by Litho 1.0 mg/mL, which was significantly different from IL-1β CM (*p* = 0.022). *CIQTNF2* expression trended towards an increased dose response in expression but did not reach the *a priori* alpha (*p* for trend = 0.061, R^2^ = 0.537). There was no effect of IL-1β or Litho on *CIQTNF2* at day 1.
Figure 3Effects of Litho on monoculture and co-cultures. Dose response curves for in Osteoblasts (**A**) and Chondrocytes (**B**) (no IL-1β CM) (no statistical analysis performed). (**C**) Co-culture cell viability across time in the OA model following treatment. (**D**) Co-culture cell viability across time in the unstimulated model (no IL-1β CM) following treatment. (**E**) IL-6 of unstimulated model following treatment. (**F**) IL-6 of OA model following treatment. (**G**) IL-8 of unstimulated model following treatment. (**H**) IL-8 of OA model following treatment. All experiments are calculated as mean data of three biological replicates, analysed in triplicate (n  =  9) are presented  ±  SEM.
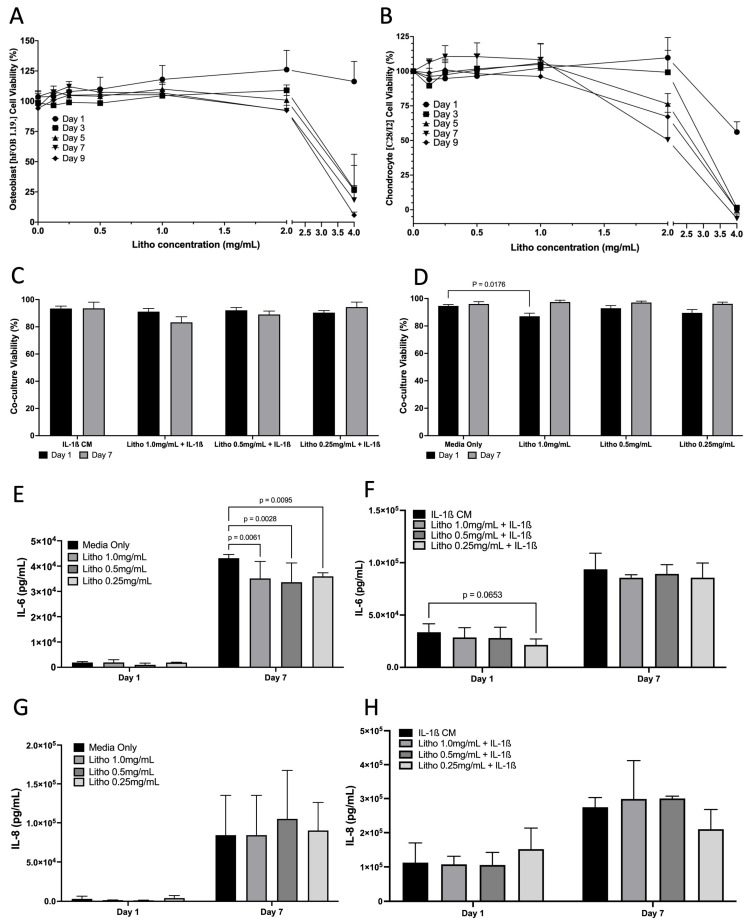

Figure 4Gene expression for the OA model. Gene expression following exposure to IL-1β CM and subsequent treatment with Litho at 0.25 mg/mL and 1.0 mg/mL. (**A**) Heat map showing relative gene expression normalised to media only across time. (**B**) Heat map showing relative gene expression following treatment. Samples were normalised to OA model (IL1-β CM). Heat map colour intensity represents fold change in gene expression. Cells with an “X” represent genes that were not expressed sufficiently for detection. (**C**) ddPCR gene expression for selected confirmatory analysis following arrays for *COL1A1*; (**D**) *IL-6*; (**E**) *CIQTNF2*. The dashed line represents *p* for Trend (*p* = 0.061); (**F**) *PTGS2*; (**G**) *TNFS10*. Data presented in copies/µL.
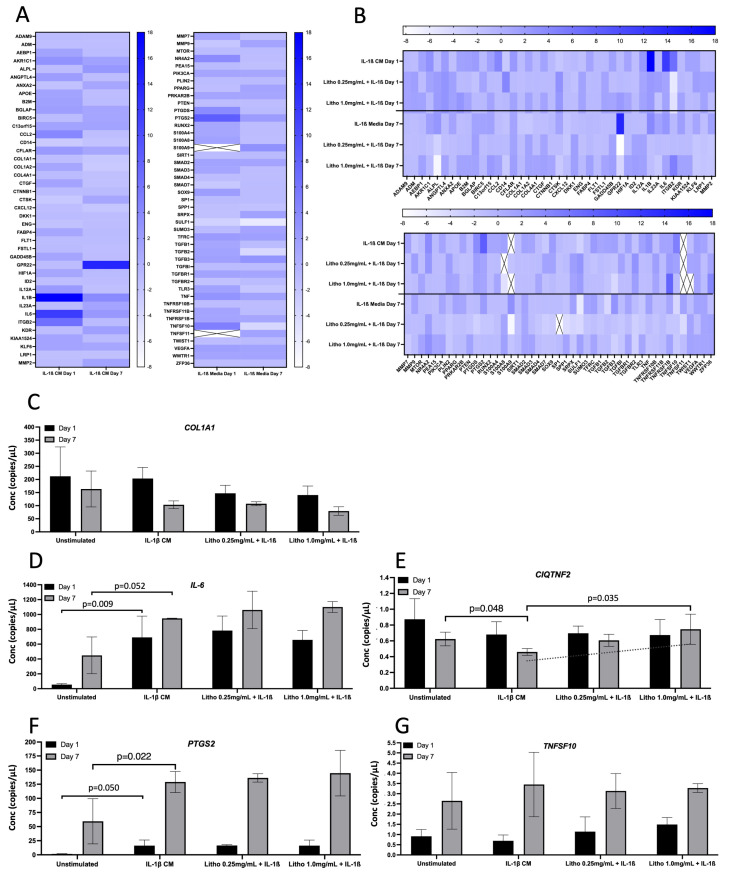



## 3. Discussion

To date, *in vitro* co-culture models of OA have mostly focused on cartilage only, often with animal tissues/cells as proxies for human disease and have not considered bone cell stimuli or cell line crosstalk [49,58,59]. Building on previous methods [60,61], the present study developed a readily accessible physiologically relevant human *in vitro* pro-inflammatory co-culture model representing OA (stimulated by IL-1β [57,58]). The co-culture model utilised cell lines that require the same media environment, was effective across at least seven days and was designed for ease of use while maintaining hFOB 1.19. and C28/I2 cell phenotypic characteristics. In addition to providing evidence of general OA inflammatory status from IL-1β stimulation [57], the present data showed confirmed gene expression of well know OA pain and inflammatory markers *IL-6* [62] and *PTGS2* [63,64]. This suggests that the present 2D human cell line co-culture model can be used as a reproducible tool to investigate both inflammation and molecular pain phenotypes in OA.

By combining these two cell lines, the model mimics, to some degree, the native *in vivo* OA condition by considering osteochondral crosstalk [59]. For example, stimulation with IL-1β can decrease viability in C28/I2 cells [65] and increase viability in hFOB 1.19. [66], thus the present model neutralized these individual physiological responses resulting in stable pooled viability and a more native OA environment. This likely occurred because as IL-1β induced apoptosis in C28/I2, it simultaneously increased the transcriptional activity of Activator Protein-1, thus modulating the impact of IL-1β on cell death in both cell lines [65,67,68]. Further, the present data showed increased levels of *COL1A1* over time which can be an indicator of dedifferentiation in chondrocytes, a common occurrence for immortalised and primary chondrocytes grown *in vitro* [69,70]. However, chondrocytes can acquire many degenerated phenotypes at the onset of OA, including a “dedifferentiated-like” phenotype that may contribute to OA progression [71]. Therefore further enhancing the physiological relevance of the model. A number of chondrocyte cell lines have been generated in an effort to overcome the dedifferentiation process however none are derived from articular cartilage making them less suitable for OA investigations. More importantly, they have been shown to be less sensitive to inflammatory stimuli [70], which is of particular importance in OA studies including the present one.

To investigate the effectiveness of the model for assessing OA related molecular mechanisms, the present study used a bioactive nutraceutical that has previously been shown to improved OA symptoms *in vivo* (Litho). As such, the RNA arrays revealed that Litho reduced the expression of key inflammatory genes, most notably *IL-1β* and *IL-6.* While these expressions were not confirmed by ddPCR, Litho did reduced extracellular IL-6 prior to treatment with IL-1β and tended towards a reduction in day 1 following IL-1β stimulation. It is possible that Litho could be affecting the inflammatory mediators at the protein level and less so at the gene expression level, but this is yet to be experimentally tested. Further, the likely mechanism for suppressed inflammatory markers in the non-stimulated condition was though the NF-κB transcriptional activity, which was reduced following exposure to *Lithothamnion* species in macrophages with lipopolysaccharide [56]. This action may have resonated, in part, from the mineral structure of *Lithothamnion* as a number of individual minerals present in Litho have been shown to negatively regulate NF-κB [72]. For example, Mg can down regulate inflammatory genes (*IL-1β*, *IL-6* and *IL-10*) and decrease NF-κB nuclear translocation and phosphorylation [73] The IL-1β stimulus used in the present study may have overpowered some of the inflammatory mediating capacity of Litho as *in vivo* studies utilising *Lithothamnion* species have shown a significant effect in mild-moderate disease, rather than severe OA [51,53,54]. Nonetheless, these findings agree somewhat with previous studies investigating *Lithothamnion* [55,56,74] and confirm that this species of Algae has anti-inflammatory properties, as evident in the present model, that warrant further mechanistic investigation. Some larger error were present in individual outcome measures; however these were not completely unexpected given the use of multiple cell lines, inflammatory stimulus and treatment with an organic material. Nonetheless, additional caution is required when interpreting these data.

Litho also reduced the expression of Prostaglandin-endoperoxide synthase 2 (*PTGS2*; also named COX-2) by ~7 fold on day 1 and maintained this reduction of >2 fold on day 7. Reduced expression of this gene signifies a potential capacity to improve the disease state as *PTGS2* can promote inflammation and oxidative stress, while inflammatory cytokines and oxidative stress-related substances increase its expression [75,76,77]. This finding is further supported by an LPS stimulated murine macrophage model that was acutely treated with *Lithothamnion* (0.5 mg/mL) and reduce *COX-2* relative expression over a 6-h period [56]. As IL-1β induces COX-2 (*PTGS2* in the present findings) in OA specific tissues [78], this is further supported by the present finding that Litho reduced the expression of *IL-1β* by ~16 fold and other data showing that *Lithothamnion* species can reduce extracellular IL-1β by ~2 fold, again in macrophages [55]. Given the nature of these markers as pain therapy targets it is possible that the mineral composition of Litho and potential delivery mechanism [79] could provide a possible mechanistic indication. As above, Mg (the second most abundant mineral present in Litho) has a growing body of evidence for its potential therapeutic role in OA [80] and has the capacity the inhibit the expression and enzymatic activity of COX-2 [81]. This is supported *in vivo*, where the use of *Lithothamnion* species as part of a nutraceutical combination reduced the use of analgesics by ~70% during a 12 week intervention [51]. However, the precise mechanisms are yet to be elucidated.

A novel finding of the current study is that of the *C1QTNF2* gene. Also known as C1q/TNF-related protein (*CTRP*
*[2]*), *C1QTNF2* is a member of a highly conserved family of proteins known mostly for their role in metabolism and as adipokines, but also for their anti-inflammatory properties [82,83]. CTRPs are paralogs of adiponectin which is a well-known adipokine that possess anti-inflammatory capacity, of which CTRP2 mostly resembles the amino acid composition [84]. Specifically, *CTRP2* expression has been shown to reduce oxidative stress, inhibit cytokine production such as TNF-α and IL-6 and has been linked to a number of inflammatory associated disease states (e.g., Coronary Artery Disease) [85]. In agreement with these roles, the present findings showed that following seven days of inflammatory exposure with IL-1β *C1QTNF2* expression was reduced and with the addition of Litho, *C1QTNF2* expression returned to untreated levels, in a dose dependent manor. These data demonstrated, for the first time, that *C1QTNF2* gene expression may have a role in OA disease state and that it may be reversed by a bioactive nutraceutical.

## 4. Materials and Methods

### 4.1. Cell Culture and Maintenance

Immortalised human foetal osteoblastic (hFOB 1.19. CRL-11372, American Type Culture Collection (ATCC) Manassas, VA, USA) and human immortalised Chondrocyte (C28/I2; Merck, Dorset, UK) cell lines were cultured in phenol-free Dulbecco’s modified Eagle’s medium (GIBCO, Paisley, UK) supplemented with 1% penicillin, streptomycin and 10% heat-inactivated foetal bovine serum (GIBCO, Paisley, UK). Cells were cultured at 37 °C in a humidified atmosphere containing 5% CO_2_. The medium was changed every three days under aseptic conditions. Passaging of confluent monolayers were carried out at 80–90% using accutase solution (Merck, UK). C28/I2 cells were discarded after passage 10 to ensure that no abnormal morphological changes occurred, as per certificate of analysis (Merck, UK).

Cells were co-cultured indirectly using transwell inserts as previously described (Figure 5) [60]. C28/I2 cells were seeded at 1 × 10^4^ cells/mL onto the apical side of Falcon^TM^ cell culture inserts (transparent PET membrane with 3 µm pores; Fisher Scientific, Cambridge, UK, 12- or 6-well size). hFOB 1.19. cells were seeded at 1 × 10^4^ into basal compartment to later form the basal layer of the indirect co-culture. Cells were incubated at 37 °C in a humidified atmosphere containing 5% CO_2_ for 48 h. To induce an osteoarthritic inflammatory disease-state an indirect co-culture model was used, which enabled the assessment of cell-cell interaction (of the two cell types) and the effect of the co-culture on growth and behaviour of both cell types. Media from both the basal layer and transwell were aspirated and replaced with IL-1β (Merck, UK) conditioned media (CM), a known OA stimulant [57,58], at 10 ng/mL (total volume = 4.5 mL per co-culture well) and incubated in the same conditions for a further 24 h. For both the apical and basal layer separately, IL-1β CM was discarded and replaced with experimental material (see below) in IL-1β CM or media only as a control. Inserts were then transferred into the appropriate basal compartment, containing experimental material plus IL-1β media (or media only as control) and hFOB 1.19 cells. Cells were harvested after 24 h of treatment (day 1). The remaining cells underwent a media change with fresh experimental media plus IL-1β (or media only) every two days and harvested at day 7. During each harvest, supernatant was collected, centrifuged at 230× *g* for 5 min and stored at −20 °C until further analysis.

### 4.2. Experimental Material

The Algae material was harvested from the calcareous cytoskeleton of *Lithothamnion* species, a member of the Corallinacea family. By utilising proprietary extraction technology (Nordic Medical Ltd., London, UK), its multimolecular complex was preserved retaining a unique porous microstructure [79,86] (referred to herein as Litho).

*Lithothamnion* species are rich in calcium (Ca), magnesium (Mg) and a variety of trace elements that are absorbed from sea-water during the organism’s life [51,87]. The application used in this experiment was recognised as safe for human consumption by the Food and Drug Administration FDA (GRAS 000028). The Red Algae extract contained ~12% Ca, ~1% Mg and measurable levels of 72 other trace minerals [74] (batch confirmed by chemical analysis, Marigot Ltd. Co Cork, Carrigaline, Ireland) and is commercially available in the UK as LithoLexal^®^ (London, UK). The experimental solution was prepared by re-suspending in phenol-free Dulbecco’s modified Eagle’s medium (DMEM; Gibco, Thermo Fisher Scientific, Waltham, MA, USA) at 37 °C for 30 min, then filter sterilised using a 0.22 µM filter. Both individual and co-culture models were treated with a range of doses 0.25–1.0 mg/mL [55,56] for either 24 h or seven days.

### 4.3. Cell Viability Assay

At the appropriate time points, media was removed from each well and replaced with 10% solution of Presto Blue cell viability reagent (Thermo Fisher Scientific, Cambridge, UK), as per manufactures instructions. All samples were run in triplicate, per biological replicate (n = 3). Fluorescence was measured with an automated microplate fluorometer (FLUOstar Omega, BMG LabTech, Aylesbury, Shropshire, UK) using an excitation wavelength of 544 nm and an emission wavelength of 590 nm. Cell viability was calculated as a percentage of the untreated negative control, with H_2_O_2_ 1 mM as the positive control.

### 4.4. Immunofluorescence

C28/I2 were stained with chondrocyte marker Aggrecan. Transwell inserts were washed (0.1% Bovine Serum Albumin (BSA) in PBS) and fixed (4% paraformaldehyde (PFA) to the apical and basal sides) at room temperature (RT) for 15 min. Inserts were washed, permeabilised (0.1% *v*/*v* Triton X-100 for 20 min), and washed twice before being blocked using 1% *w*/*v* BSA for 1 h at RT. Cells were stained with Aggrecan monoclonal primary antibody (1:200) (Thermo Scientific, Cambridge, UK) for 3 h at 4 °C. Before incubating with secondary antibody, Goat anti-mouse IgG Alexa Fluro 488 (Ex 490 nm/Em 525 nm) (1:500) (Thermo Scientific, Cambridge, UK) and Phalloidin Alexa Fluor 594 (Ex 590 nm/Em 618 nm) staining F-actin (1:400) (Thermo Scientific, Cambridge, UK) for 1 h at RT. Inserts were then mounted with Vectashield mounting medium Dapi (Ex 360 nm/Em 460 nm) (Vector Laboratories, Peterborough, UK) and imaged Zeiss LSM980 confocal microscope (Carl Zeiss AG, Oberkochen, Germany).

### 4.5. Von Kossa Staining

hFOB 1.19. cells were assayed with Von Kossa (150687, Abcam, Cambridge, UK) to identify calcium specific deposition as per manufactures instructions. Briefly, cells were washed twice with PBS, fixed with 4% PFA for 30 min at RT. Cells were washed twice with PBS and incubated in silver nitrate solution (5%) for 30–60 min under ultraviolet light (Analytik Jena UV Lamp, Fisher Scientific, Cambridge, UK), then washed again. Unreacted silver nitrate was removed by adding 5% sodium thiosulfate for 5 min. Nuclear factor red staining solution was then added and incubated for 5 min and washed in PBS three times. Cells were imaged by light microscopy (EVOS XL, Thermo Scientific, Cambridge, UK). All samples were run in triplicate per biological replicate (n = 3).

### 4.6. Alizarin Red-S Assay

Extra-articular mineralisation and non-specific calcium deposition in hFOB 1.19. were determined by Alizarin red-S assay staining (qualitative) followed by extraction of the stained calcium-rich nodules (deposits) for quantification by spectrophotometry. Cells were washed twice with PBS before fixing with 4% PFA solution and incubated at RT for 20 min. Cells were washed twice, stained with 40 mM Alizarin red-S (pH 4.2) (Sigma-Aldrich, St. Louis, MO, USA, A5533) and incubated on a shaker for 85 min (protected from light). Cells were washed twice with PBS for 5 min each with fresh PBS added for imaging, using a light microscope (EVOS XL, Thermo Scientific, Cambridge, UK). For quantification, cells were lysed with 10% acetic acid for 30 min at RT, transferred into 1.5 mL tubes, heated at 85 °C for 10 min, centrifuged at 20,000× *g* for 15 min at 4 °C. The lysed cells were then neutralized with 10% ammonium hydroxide for assaying. Absorbance was read at optical density (OD) 405 nm (FLUOstar Omega, BMG LabTech, Aylesbury, Shropshire, UK). The concentration of alizarin red-S (µg/L) was determined according to the linear regression equation derived from the standard curve. All samples were run in triplicate per biological replicate (n = 3).

### 4.7. Alkaline Phosphatase Assay

Alkaline Phosphatase (ALP; ab83371, Abcam, Cambridge, UK) was used as a marker of osteogenesis and a characteristic indicator of osteoblastic growth and was performed as per manufactures instructions. Briefly, 4-methylumbelliferyl (4-MU) phosphate disodium salt (MUP; 50 μM) reaction mix was added to each well of a 96 well plate. Cell culture supernatant was diluted 1:15 in PBS before adding to reaction mix and incubated for 30 min at RT, protected from light. Stop solution was added and fluorescence was measured spectrophotometrically (FLUOstar Omega, BMG LabTech, Aylesbury, Shropshire, UK), excitation 355 nm and emission 460 nm. Generated 4-MU was calculated according to the linear regression equation derived from the standard curve. All samples were run in triplicate per biological replicate (n = 3).

### 4.8. Pro-Inflammatory Assessment

The concentration of (pro-)inflammatory mediators released into the medium was measured via Enzyme-Linked Immunosorbent Assay (ELISA). IL-1β was used as a positive (pro)-inflammatory control at 10 ng/mL added to both the apical and basal side of the culture. Cell culture supernatant was collected at day 1 and 7 after treatment and analysed for IL-8 (Cat no. DY208) and IL-6 (Cat no. DY206) using DuoSet kits from R&D systems (Bio-techne, Abingdon, UK) according to the manufacturer’s instructions. All samples were run in triplicate per biological replicate (n = 3) and absorbance was measured at 450 nm with background correction at 570 nm. Extrapolation of IL-6 and IL-8 concentration was carried out using a four-parameter logistic curve using Graphpad PRISM (Version 10.0.3).

### 4.9. Total RNA Isolation, Quantification and Reverse Transcription PCR

Following treatments, total RNA was extracted from co-culture using the RNeasy mini kit (Qiagen Ltd., Surrey, UK) as per manufactures instructions. The amount and quality of RNA [optical density (OD) ratio 260/280 > 1.8, OD ratio 230/280 > 1.7] was measured using a Nanophotometer (IMPLEN, Munich, Germany). Total RNA (1000 ng/μL) was converted to cDNA using the iScript™ gDNA Clear cDNA Synthesis Kit (Bio-Rad Laboratories Ltd., Watford, UK) as per manufactures instructions with a final reaction volume of 20 µL. The reaction volume was incubated on the T100 thermal cycler (Bio-Rad Laboratories Ltd., Watford, UK) for 5 min at 25 °C, then at 46 °C for 20 min and finally 95 °C for 1 min. cDNA (1:10 dilution) was added to predesigned Human Osteoarthritis H96 qPCR Arrays in a 96-well format using SsoAdvanced Universal SYBR Green Supermix (Bio-Rad Laboratories Ltd., Watford, UK) in duplicate as per manufactures instructions and using the Bio-Rad CFX connect (Bio-Rad Laboratories Ltd., Watford, UK). Gene expression data was calculated using the ΔΔCq method. Data was normalized to three reference genes (HPRT1, GAPDH, TBP) using Bio-Rad CFX Maestro Data Analysis software (Version 2.0; Bio-Rad Laboratories Ltd., Watford, UK). Litho samples were then compared to OA model (IL-1β CM) for changes in gene expression. Results were plotted in GraphPad prism software (Version 10.0.3) as Fold Change using mean ± SEM.

Informed by array data and literature review, a panel of genes were selected for confirmatory analysis by droplet digital PCR (ddPCR). All primers and reagents were purchased from Bio-Rad (Bio-Rad Laboratories Ltd., Watford, UK). The ddPCR assays were performed on the same cDNA samples previously mentioned. For ddPCR, the final reaction volume was 20 μL, which included cDNA mix, 2X ddPCR Supermix for probes (no dUTP), 20x target probe FAM, 20x target probe HEX and RNase free H_2_O. All primer probes were predesigned and purchased from Bio-Rad; *COL1A1* (HEX), *C1qTNF2* (FAM), *IL-6* (FAM), *PTGS2* (HEX), *CCL2* (FAM), *TNFSF10* (HEX), *TGFB3* (HEX), *PTDGS* (FAM). The droplet generation and reading were performed according to the manufacturer’s instructions. Briefly, 70 μL of probe droplet generation oil was added. The water-in-oil droplet emulsion was prepared using the QX200 Droplet Generator (Bio-Rad). The amplification step was performed using 40 μL of emulsion in a C1000 thermal cycler (Bio-Rad Laboratories Ltd., Watford, UK). The ddPCR reactions were analysed using the QX200 Droplet Reader and QuantaSoft software version 1.7.4 (both from Bio-Rad).

### 4.10. Statistical Analysis

Data are presented as means ± standard error of the mean (SEM), fold change and copies/μL were applicable. Shapiro-Wilk test was used to test for normality of each data set. For normally distributed characteristics and ELISAs, either one-way analysis of variance (ANOVA; ARZ and ALP), *t*-tests (*COL1A1* and untreated viability) or two-way ANOVA (IL-6, IL-8 and treated cell viability) were performed. For normally distributed gene expression data, *t*-tests were performed for unstimulated vs. unstimulated (IL-1β CM) (*COL1A1*, *IL-6*, *CIQTNF2* and *PTGS2* (day 7)) or for nonparametric data Mann-Whitney U was performed (*PTGS2* (day 1)). For treatment comparisons of gene expression data, either two-way ANOVA (*COL1A1*, *IL-6*, *CIQTNF2*, *PTGS2* (day 7) and *TNFSF10*) with Tukey or Dunnetts post hoc was applied, unless otherwise stated. For nonparametric data Kruskal-Wallis was used (PTGS2 (day 1) and CIQTNF2 (day 1)). Gene expression array data was presented as normalised fold change. All null-hypothesis tests were performed in Graphpad PRISM (Version 10.0.3) with alpha set at *p* < 0.05.

## 5. Conclusions

This study presents for the first time a replicable 2D co-culture model of inflammatory OA for investigations of bioactive nutraceuticals. The model showed that one such nutraceutical, Litho, reduced the expression of inflammatory and pain related genes, and demonstrated for the first time altered expression of *C1QTNF2* (*CTRP2*) in a human OA model. The precise role of *C1QTNF2* in OA is unknown and the potential for *Lithothamnion* species to modify its expression warrants further investigation. These data and the present model can be used to further investigate if *Lithothamnion* species and other bioactive nutraceuticals can be used in tandem with pharmaceutical treatments to act synergistically. This could show the potentially for bioactive nutraceuticals to reduce the dosages of harmful often-synthetic pharmaceuticals and if they could potentially lead to reduced incidence of adverse side effects currently being experienced by OA patients during treatment.

## Figures and Tables

**Figure 1 pharmaceuticals-18-00315-f001:**
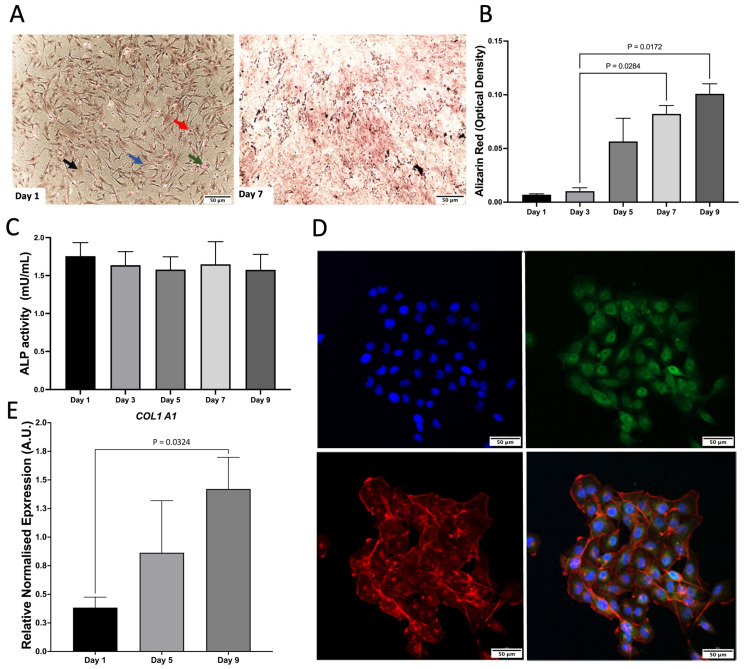
Characterisation of monocultures. (**A**) Von Kossa staining of hFOB 1.19. on day 1 and 7. Red arrow, calcium mass deposits stained black. Blue arrow, calcium dispensed deposits stained grey; Green arrow, nuclei stained red; Black arrow, cytoplasm stained light pink. (**B**) Alizarin Red (ARZ) quantification showing osteoblast cell calcium/mineral deposition across days 1–9 of cell growth, n = 3 with assays performed in triplicate. The data is presented as the mean ± SEM. (**C**) Alkaline phosphatase activity (ALP) for hFOB 1.19, n = 3 with assays performed in triplicate (*p* = 0.972). The data is presented as the mean ± SEM. (**D**) Confocal microscopy images of aggrecan (day 1). Cells were stained with aggrecan (green), nuclear stain dapi (blue), and phalloidin to stain f-actin (red). Scale bar: 50 μm. Images are representative of 3 biological replicates. (**E**) *COL1A1* gene expression of C28/I2 compared to day 1. This data represents n = 3 biological replicates.

**Figure 2 pharmaceuticals-18-00315-f002:**
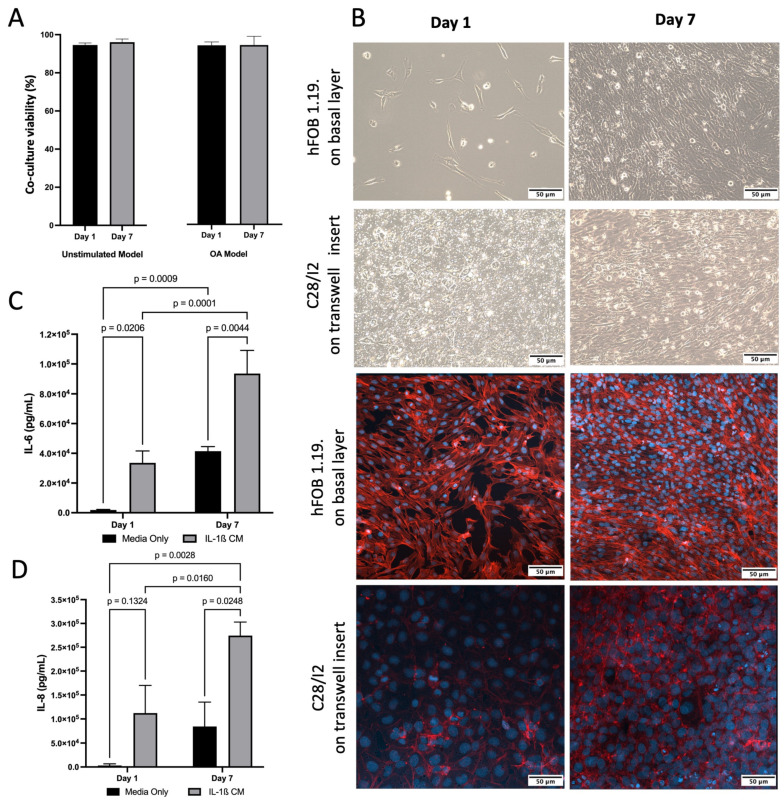
Characterisation of inflamed co-culture model. (**A**) Co-culture cell viability and cell size across time following induction of inflammatory state via IL-1β CM (OA Model, *p* > 0.05) or media only (Unstimulated Model, *p* > 0.05). (**B**) (**Top**) Inverted light microscopy images of hFOB 1.19 cells seeded at 1 × 10^4^ in media only on the basal layer at day 1 and day 7. C28/I2 cells seeded at 1 × 10^4^ on transwell inserts in media only day 1 and day 7. (**Bottom**) Confocal Microscopy images stained for f-actin (red—phalloidin) and nucleus (blue—dapi) showing that cells grew on the transwell and did not traverse the membrane (C28/I2), and hFOB 1.19 cells on basal layer. Images scaled at 50 μm, and representative of three biological replicates. (**C**) Elevated concentrations of IL-6 following induction of inflammatory state via IL-1β CM, compared to media only on day 1 and day 7. Mean data of three biological replicates, analysed in triplicate (n  =  9) are presented  ±  SEM. (**D**) Elevated concentrations of IL-8 following induction of inflammatory state via IL-1β CM, compared to media only on day 1 and day 7. Mean data of three biological replicates, analysed in triplicate (n  =  9) are presented  ±  SEM.

**Figure 5 pharmaceuticals-18-00315-f005:**
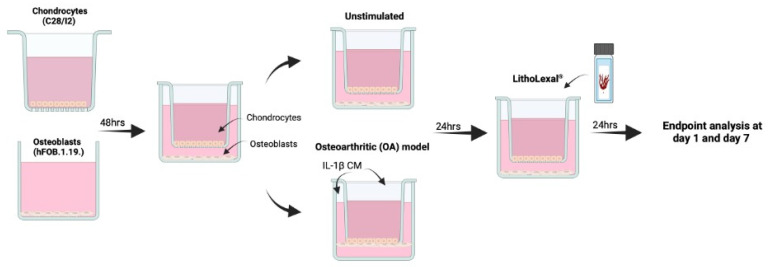
Schematic of co-culture OA model. hFOB 1.19 and C28/I2 cells were co-cultured indirectly using transwell inserts. C28/I2 cells were seeded at 1 × 10^4^ cells/mL onto the apical side of cell culture inserts and hFOB 1.19. cells were seeded at 1 × 10^4^ into basal compartment of the transwell system. Both cell lines were left for 48 h to adhere after which the transwell insert containing C28/I2 cells was added into the basal compartment, with hFOB 1.19. cells seeded on the base. To induce an OA inflammatory-like state, media was aspirated and replaced with IL-1β conditioned media (CM) at 10 ng/mL for 24 h prior to treatment. Created in BioRender. IVTG, S. (2025) https://BioRender.com/j70y844 (last accessed on 9 February 2025).

## Data Availability

Data can be made available upon reasonable request.

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
