# Peer review of "Red Algae Alters Expression of Inflammatory Pathways in an Osteoarthritis In Vitro Co-Culture"

_pharmaceuticals, 2025, doi:10.3390/ph18030315_

Round 1

Reviewer 1 Report

Comments and Suggestions for Authors

In this manuscript by Heffernan et al., the authors explore the effect of red algae in a chondrocyte-osteoblast co-culture model. This study is exploratory in nature and appears to be in its preliminary stages. Below are my comments:

The title, “---in a human in vitro osteoarthritis co-culture”, does not provide a clear understanding of the study. What do the authors mean by “in vitro osteoarthritis co-culture”? Please clarify.

Several sentences in the manuscript require grammatical corrections and improved structure. For instance: Line 244: “For nonparametric data, either Kruskal-Wallis (PTGS2(day 1), CIQTNF2(day 1)).” This sentence is incomplete and needs revision for clarity.

Figure legend 2, what do the author mean by 3 biological replicates? These experiments are done with the same cell line, not primary cells isolated from different donors.

The authors show an increased level of COL1A1 in Figure 2E. An increase in type 1 collagen is typically indicative of the dedifferentiation of C28/I2 cells into a fibroblast-like phenotype. Please elaborate on this observation and its implications.

In Figure 3A, the authors report no loss in cell viability. However, this finding is intriguing because several studies have shown increased cell death in chondrocytes stimulated with a similar concentration of IL-1β within 24 hours (e.g., PMID: 36133743, 35026522). The authors should address this discrepancy.

This paper does not investigate Litho's mechanism of action. A discussion of potential pathways or hypotheses would strengthen the manuscript.

Comments on the Quality of English Language

Several sentences in the manuscript require grammatical corrections and improved structure. For instance: Line 244: “For nonparametric data, either Kruskal-Wallis (PTGS2(day 1), CIQTNF2(day 1)).” This sentence is incomplete and needs revision for clarity.

Author Response

Comments and Suggestions for Authors

In this manuscript by Heffernan et al., the authors explore the effect of red algae in a chondrocyte-osteoblast co-culture model. This study is exploratory in nature and appears to be in its preliminary stages. Below are my comments:

The title, “---in a human in vitro osteoarthritis co-culture”, does not provide a clear understanding of the study. What do the authors mean by “in vitro osteoarthritis co-culture”? Please clarify.

Thank you for your comment. To clarify the terms in the title, “in vitro” refers to the experimental method used to explore the research question, “osteoarthritis” refers to the condition that the model was designed to investigate and “co-culture” described the model utilised.

However, to help the reviewer (and readers) interpret the title clearer we have amended the text. The title now reads,

‘Red Algae alters expression of inflammatory pathways in an osteoarthritis in vitro co-culture’

Several sentences in the manuscript require grammatical corrections and improved structure. For instance: Line 244: “For nonparametric data, either Kruskal-Wallis (PTGS2(day 1), CIQTNF2(day 1)).” This sentence is incomplete and needs revision for clarity.

We thank the reviewer for pointing out this error. The article has been further reviewed and minor changes have been implemented throughout the improve grammatical oversight and sentence structures. The error mentioned has now been rectified and reads;

‘For nonparametric data Kruskal-Wallis was used (PTGS2(day 1) and CIQTNF2(day 1)).’

Figure legend 2, what do the author mean by 3 biological replicates? These experiments are done with the same cell line, not primary cells isolated from different donors.

We thank the reviewer for their comment. The authors classify biological replicates as parallel measurements of biologically distinct samples that capture random biological variation, which may itself be a subject of study or a source of noise as per https://www.nature.com/documents/Biological_and_technical_replicates_guidelines.pdf. The authors would like to clarify that biological replicates do not need to be primary cells, but can refer to a number of different in vitro techniques.

The authors show an increased level of COL1A1 in Figure 2E. An increase in type 1 collagen is typically indicative of the dedifferentiation of C28/I2 cells into a fibroblast-like phenotype. Please elaborate on this observation and its implications.

Thank you for this additional discussion point. The authors are in agreement with the reviewer that chondrocytes are known to dedifferentiate during in vitro culture over time and so have added the below text to the discussion;

“Further, the present data showed an increased levels of COL1A1 over time which can be an indicator of dedifferentiation in chondrocytes, a common occurrence for immortalised and primary chondrocytes grown in vitro1,2. However, chondrocytes can acquire many degenerated phenotypes at the onset of OA, including a “dedifferentiated-like” phenotype that may contribute to OA progression3. Therefore further enhancing the physiological relevance of the model. A number of chondrocyte cell lines have been generated in an effort to overcome the dedifferentiation process however they are not derived from articular cartilage making them less suitable for OA investigations and more importantly they have been shown to be less sensitive to inflammatory stimuli2 which is of particular importance in OA studies including the present one.”

In Figure 3A, the authors report no loss in cell viability. However, this finding is intriguing because several studies have shown increased cell death in chondrocytes stimulated with a similar concentration of IL-1β within 24 hours (e.g., PMID: 36133743, 35026522). The authors should address this discrepancy.

Thank you for your comment, your interest and the included citations. This is an interesting line of enquiry that the authors will follow up experimentally at a later date, however we have included additional text in the discussion to address this point, which now reads;

“By combining these two cell lines, the model mimics, to some degree, the native in vivo OA condition by considering osteochondral crosstalk4. For example, stimulation with IL-1ß can decrease viability in C28/I2 cells5 and increase viability in hFOB 1.19.6, thus the present model neutralized these individual physiological response resulting in stable pooled viability and a more native OA environment. This likely occurred as IL-1ß induced apoptosis in C28/I2, it simultaneously increased the transcriptional activity of Activator Protein-1, thus modulating the impact of IL-1ß on cell death in both cell lines5,7,8.”

This paper does not investigate Litho's mechanism of action. A discussion of potential pathways or hypotheses would strengthen the manuscript.

Thank you for your comment. We have included more discussion of potential pathways throughout. The text now reads;

“It is possible that Litho could be affecting the inflammatory mediators at the protein level and less so at gene expression, but this is yet to be experimentally tested. Further, the likely mechanism for suppressed inflammatory markers in the non-stimulated condition was though the NF-kB transcriptional activity, which was reduced with exposure to Lithothamnion species in macrophage cells with lipopolysaccharide9. This action may have resonated in part from the mineral structure of Litho as a number on individual minerals present in Litho have been shown to negatively regulate NF-kB10. For example, Mg has been shown to down regulate inflammatory genes (IL-1β, IL-6 and IL-10) and decrease NF-kB nuclear translocation and phosphorylation11

“As IL-1ß induces COX-2 (PTGS2 in the present findings) in OA specific tissues12, this is further supported the present data showing that Litho reduced the expression of IL-1ß by ~16 fold and other data that Lithothamnion species can reduce extracellular IL-1ß by ~2 fold, again in macrophages13. Given the nature of these markers as pain therapy targets it is possible that the mineral composition of Litho and potential delivery mechanism14 could provide a possible mechanistic indication. As above, Mg (the second most abundant mineral in Litho) has a growing body of evidence for its potential therapeutic role in OA15 and has the capacity the inhibit the expression and enzymatic activity of COX-216. This is further supported in vivo, in that the use of Lithothamnion species as part of a nutraceutical combination reduced the use of analgesics by ~70% during a 12 week intervention17. However, the precise mechanisms of action are yet to be elucidated.

Comments on the Quality of English Language

Several sentences in the manuscript require grammatical corrections and improved structure. For instance: Line 244: “For nonparametric data, either Kruskal-Wallis (PTGS2(day 1), CIQTNF2(day 1)).” This sentence is incomplete and needs revision for clarity.

Thank you for your comment, this has now been addressed.

Supporting references (some apply to other reviewer's comments)

  1. Duan L, Ma B, Liang Y, et al. Cytokine networking of chondrocyte dedifferentiation in vitro and its implications for cell-based cartilage therapy. American Journal of Translational Research. 2015;7(2):194-208.
  2. Piñeiro-Ramil M, Sanjurjo-Rodríguez C, Rodríguez-Fernández S, et al. Generation of human immortalized chondrocytes from osteoarthritic and healthy cartilage : a new tool for cartilage pathophysiology studies. Bone and Joint Research. 2023;12(1):46-57. doi: 10.1302/2046-3758.121.Bjr-2022-0207.R1.
  3. Charlier E, Deroyer C, Ciregia F, et al. Chondrocyte dedifferentiation and osteoarthritis (OA). Biochemical Pharmacology. 2019;165:49-65. doi: https://doi.org/10.1016/j.bcp.2019.02.036.
  4. Jiang A, Xu P, Sun S, et al. Cellular alterations and crosstalk in the osteochondral joint in osteoarthritis and promising therapeutic strategies. Connective Tissue Research. 2021;62(6):709-719. doi: 10.1080/03008207.2020.1870969.
  5. Meng Y, Yin D, Qiu S, Zhang X. Abrine promotes cell proliferation and inhibits apoptosis of interleukin-1β-stimulated chondrocytes via PIM2/VEGF signalling in osteoarthritis. Phytomedicine. 2022;96:153906. doi: 10.1016/j.phymed.2021.153906.
  6. Ying H, Li Q, Zhao C. Interleukin 1β and tumor necrosis factor α promote hFOB1.19 cell viability via activating AP1. American Journal of Translational Research. 2016;8(5):2411-2418.
  7. Zhou Y, Zhou B, Pache L, et al. Metascape provides a biologist-oriented resource for the analysis of systems-level datasets. Nature Communications. 2019;10(1):1523. doi: 10.1038/s41467-019-09234-6.
  8. Garces de Los Fayos Alonso I, Liang HC, Turner SD, Lagger S, Merkel O, Kenner L. The Role of Activator Protein-1 (AP-1) Family Members in CD30-Positive Lymphomas. Cancers (Basel). 2018;10(4). doi: 10.3390/cancers10040093.
  9. O'Gorman DM, O'Carroll C, Carmody RJ. Evidence that marine-derived, multi-mineral, Aquamin inhibits the NF-κB signaling pathway in vitro. Phytother Res. 2012;26(4):630-632. doi: 10.1002/ptr.3601.
  10. Weyh C, Krüger K, Peeling P, Castell L. The Role of Minerals in the Optimal Functioning of the Immune System. Nutrients. 2022;14(3). doi: 10.3390/nu14030644.
  11. Hu T, Xu H, Wang C, Qin H, An Z. Magnesium enhances the chondrogenic differentiation of mesenchymal stem cells by inhibiting activated macrophage-induced inflammation. Scientific Reports. 2018;8(1):3406. doi: 10.1038/s41598-018-21783-2.
  12. Mastbergen SC, Lafeber FP, Bijlsma JW. Selective COX-2 inhibition prevents proinflammatory cytokine-induced cartilage damage. Rheumatology. 2002;41(7):801-808. doi: 10.1093/rheumatology/41.7.801.
  13. Ryan S, O'Gorman DM, Nolan YM. Evidence that the marine-derived multi-mineral Aquamin has anti-inflammatory effects on cortical glial-enriched cultures. Phytotherapy Research. 2011;25(5):765-767. doi: 10.1002/ptr.3309.
  14. Eriksen EF, Lech O, Nakama GY, O'Gorman DM. Disease-Modifying Adjunctive Therapy (DMAT) in Osteoarthritis-The Biological Effects of a Multi-Mineral Complex, LithoLexal(®) Joint-A Review. Clin Pract. 2021;11(4):901-913. doi: 10.3390/clinpract11040104.
  15. Li Y, Yue J, Yang C. Unraveling the role of Mg++ in osteoarthritis. Life Sciences. 2016;147:24-29. doi: https://doi.org/10.1016/j.lfs.2016.01.029.
  16. Guan P-P, Jia J-F, Wang P. Dietary magnesium ions block the effects of nutrient deficiency on inducing the formation of LC3B autophagosomes and disrupting the proteolysis of autolysosomes to degrade β-amyloid protein by activating cyclooloxygenase-2 at tyrosine 385. Journal of Functional Foods. 2021;83:104527. doi: https://doi.org/10.1016/j.jff.2021.104527.
  17. Heffernan SM, McCarthy C, Eustace S, FitzPatrick RE, Delahunt E, De Vito G. Mineral rich algae with pine bark improved pain, physical function and analgesic use in mild-knee joint osteoarthritis, compared to Glucosamine: a randomized controlled pilot trial. Complementary Therapies in Medicine. 2020;50:102349. doi: 10.1016/j.ctim.2020.102349.
  18. Bartolotti I, Roseti L, Petretta M, Grigolo B, Desando G. A Roadmap of In Vitro Models in Osteoarthritis: A Focus on Their Biological Relevance in Regenerative Medicine. Journal of Clinical Medicine. 2021;10(9). doi: 10.3390/jcm10091920.
  19. Aslam MN, Kreider JM, Paruchuri T, et al. A mineral-rich extract from the red marine algae Lithothamnion calcareum preserves bone structure and function in female mice on a Western-style diet. Calcified Tissue International. 2010;86(4):313-324. doi: 10.1007/s00223-010-9340-9.
  20. Frestedt JL, Walsh M, Kuskowski MA, Zenk JL. A natural mineral supplement provides relief from knee osteoarthritis symptoms: a randomized controlled pilot trial. Nutrition Journal. 2008;7:9. doi: 10.1186/1475-2891-7-9.
  21. Frestedt JL, Kuskowski MA, Zenk JL. A natural seaweed derived mineral supplement (Aquamin F) for knee osteoarthritis: a randomised, placebo controlled pilot study. Nutrition Journal. 2009;8:7. doi: 10.1186/1475-2891-8-7.
  22. Taylor SC, Laperriere G, Germain H. Droplet Digital PCR versus qPCR for gene expression analysis with low abundant targets: from variable nonsense to publication quality data. Scientific Reports. 2017;7(1):2409. doi: 10.1038/s41598-017-02217-x.
  23. Salgado C, Jordan O, Allémann E. Osteoarthritis In Vitro Models: Applications and Implications in Development of Intra-Articular Drug Delivery Systems. Pharmaceutics. 2021;13(1). doi: 10.3390/pharmaceutics13010060.
  24. Heffernan SM, Conway GE, McCarthy C, et al. Inflammatory markers in early knee joint osteoarthritis differ from well-matched controls and are associated with consistent, rather than intermittent knee pain. Knee. 2024;51:189-198. doi: 10.1016/j.knee.2024.09.003.
  25. Molnar V, Matišić V, Kodvanj I, et al. Cytokines and Chemokines Involved in Osteoarthritis Pathogenesis. Internal Journal of Molecular Science. 2021;22(17). doi: 10.3390/ijms22179208.

Reviewer 2 Report

Comments and Suggestions for Authors

Comments for the Authors

In this review article, the authors have explored an intriguing subject that has been somewhat overlooked in recent years, despite Red Algae extract alters expression of inflammatory pathways 2 in a human in vitro osteoarthritis co-culture. Their focus on the recent advancements in this area offers valuable insights into Dementia and more. Considering the limited availability of such articles, this manuscript holds considerable potential for publication in Pharmaceuticals. However, further polishing is required to meet the journal's standards. With its concise content, this article promises to offer valuable insights to scientists. After undergoing minor revisions, I endorse the publication of this manuscript in Pharmaceuticals, where it can contribute significantly to the field.

1.    This manuscript is well-written; however, it requires the better version of english modification need for comprehensive understanding.

2.    In the line 100, 101 Authors should provide comment on modeling osteoarthritic inflammatory disease states how it makes difference in trans well inserts facilitate the study of indirect co-culture systems.

3.    Authors should explain about how IL-1ß influence  inducing an osteoarthritic inflammatory response in the co-culture system, will it any modification of experimental treatments.

4.     In the line 220, how it changes, by utilizing of qPCR and ddPCR complement each other for analyzing gene expression changes in the osteoarthritic co-culture model.

5.    In the line 214, when when conducting relative quantification in the osteoarthritic model, what is normalizing gene expression data to reference genes (HPRT1, GAPDH, TBP) need more explanation.

6.     In the line 278, in the co-culture model, when increases in IL-6 and IL-8 levels over time validate the effectso f IL-1ß? need more explanation.

7.     In the line 273, What is the significance of maintaining high cell viability (>93%) and consistent cell size (~16.5µm) throughout the 7-day experimental period in the context of the osteoarthritic inflammation study?

Comments on the Quality of English Language

Comments for the Authors

In this review article, the authors have explored an intriguing subject that has been somewhat overlooked in recent years, despite Red Algae extract alters expression of inflammatory pathways 2 in a human in vitro osteoarthritis co-culture. Their focus on the recent advancements in this area offers valuable insights into Dementia and more. Considering the limited availability of such articles, this manuscript holds considerable potential for publication in Pharmaceuticals. However, further polishing is required to meet the journal's standards. With its concise content, this article promises to offer valuable insights to scientists. After undergoing minor revisions, I endorse the publication of this manuscript in Pharmaceuticals, where it can contribute significantly to the field.

1.    This manuscript is well-written; however, it requires the better version of english modification need for comprehensive understanding.

2.    In the line 100, 101 Authors should provide comment on modeling osteoarthritic inflammatory disease states how it makes difference in trans well inserts facilitate the study of indirect co-culture systems.

3.    Authors should explain about how IL-1ß influence  inducing an osteoarthritic inflammatory response in the co-culture system, will it any modification of experimental treatments.

4.     In the line 220, how it changes, by utilizing of qPCR and ddPCR complement each other for analyzing gene expression changes in the osteoarthritic co-culture model.

5.    In the line 214, when when conducting relative quantification in the osteoarthritic model, what is normalizing gene expression data to reference genes (HPRT1, GAPDH, TBP) need more explanation.

6.     In the line 278, in the co-culture model, when increases in IL-6 and IL-8 levels over time validate the effectso f IL-1ß? need more explanation.

7.     In the line 273, What is the significance of maintaining high cell viability (>93%) and consistent cell size (~16.5µm) throughout the 7-day experimental period in the context of the osteoarthritic inflammation study?

Author Response

Comments and Suggestions for Authors

Comments for the Authors

In this review article, the authors have explored an intriguing subject that has been somewhat overlooked in recent years, despite Red Algae extract alters expression of inflammatory pathways 2 in a human in vitro osteoarthritis co-culture. Their focus on the recent advancements in this area offers valuable insights into Dementia and more. Considering the limited availability of such articles, this manuscript holds considerable potential for publication in Pharmaceuticals. However, further polishing is required to meet the journal's standards. With its concise content, this article promises to offer valuable insights to scientists. After undergoing minor revisions, I endorse the publication of this manuscript in Pharmaceuticals, where it can contribute significantly to the field.

  1. This manuscript is well-written; however, it requires the better version of english modification need for comprehensive understanding.

Thank you for your comment. The article has been further reviewed, and minor changes have been implemented throughout improving grammatical oversights and sentence structures.

  1. In the line 100, 101 Authors should provide comment on modeling osteoarthritic inflammatory disease states how it makes difference in trans well inserts facilitate the study of indirect co-culture systems.

The Authors thank the Reviewer for this important discussion point and the following has been added to the text to clarify for both the Reviewer and the readers:. The text now reads;

“To induce an osteoarthritic inflammatory disease state an indirect co-culture model was utilised which enabled the assessment of cell-cell interaction of different cell types and the effect of co-culture on growth and behaviour of both cell types. Media from both the basal layer and transwell insert was aspirated and replaced with known OA stimulant IL-1ß (Merck, UK) conditioned media (CM)58,59 at 10ng/mL (total volume = 4.5 mL per co-culture well) and incubated with the same conditions for a further 24 hrs.”

  1. Authors should explain about how IL-1ß influence  inducing an osteoarthritic inflammatory response in the co-culture system, will it any modification of experimental treatments.

Thank you for the opportunity to explain the influence of IL-1B further.

IL-1ß is among the most used inflammatory stimulators of OA because it is evident and detectible throughout the course of the disease (early to severe OA), it disrupts collagen synthesis and induces apoptosis in OA cells18, all typical markers of the disease. Our experimental protocol was designed (creation of an inflammatory co-culture model) to ask exactly the question we interpret the reviewer to have asked. Our data has shown that the experimental material was able to ameliorate some of the effect of the inflammatory stimulus on gene expression of OA related markers. Thus showing that any modification was beneficial for improving the OA status of the model.

Given the nature of the organic material, as a result of the processing to render Litho soluble i.e. milling, it is unlikely that IL-1ß itself would have any impact of the structure of the compound or its potential to impact inflammatory status of the model. This hypothesis is supported as this material has been used in previous inflammatory models/studies and been shown effective 9,13,19-21 as was the case in the present results. However, the direct impact on the properties of the material is currently unknown and given the reviewers comment the authors will consider this for future work.

  1. In the line 220, how it changes, by utilizing of qPCR and ddPCR complement each other for analyzing gene expression changes in the osteoarthritic co-culture model.

Thank you for your comment. We are not quite sure what the exact request was from the reviewer. However, we have attempted to interpret the reviewers comment which is detailed below:

qPCR was used for the array plate i.e. multiplex panel of genes intended to explore potential pathways of interest. It is common practice that qPCR data is accompanied by confirmatory qPCR, therefore the authors opted to use ddPCR which is better for detecting low-expressing targets and has higher precision and sensitivity, whereas qPCR is better for detecting larger differences in expression22

  1. In the line 214, when when conducting relative quantification in the osteoarthritic model, what is normalizing gene expression data to reference genes (HPRT1, GAPDH, TBP) need more explanation.

Thank you for your comment. The text has been amended to clarify the type of analysis.

“Gene expression data was calculated using the ΔΔCq method. Data was normalized to three reference genes (HPRT1, GAPDH, TBP) using Bio-Rad CFX Maestro Data Analysis software (Bio-Rad Laboratories Ltd, Watford, UK). Litho samples were then compared to OA model (IL-1ß CM) for changes in gene expression. Results were plotted in GraphPad prism software (Version 10.0.3) as Fold Change using mean ± SEM.”

  1. In the line 278, in the co-culture model, when increases in IL-6 and IL-8 levels over time validate the effectso f IL-1ß? need more explanation.

Thank you for your comment. The increases in IL-6 and IL-8 were confirmation that the co-culture model, with IL-1ß, induced inflammation thus achieving the aim of the experimental design. However given the authors comment the text has been amended as such;

“To induce inflammation, thus simulating an osteoarthritic-like scenario (OA model)23, IL-1ß was added to the co-culture at 10 ng/mL, viability (> 93%) and cell size remained (~16.5 μm) consistent with no significant drop in viability over 7 days (Figure 3 A). The presence of inflammation was confirmed by IL-6 and IL-8 ELISAs, markers known to be present and often used in OA experimental designs23,24. There was a significant increase in IL-6 between day 1 and day 7.”

  1.  In the line 273, What is the significance of maintaining high cell viability (>93%) and consistent cell size (~16.5µm) throughout the 7-day experimental period in the context of the osteoarthritic inflammation study?

Thank you for your comment. The inclusion of this details was to show that across the experimental period, with and without IL-1ß the cells maintained a healthy phenotype. This detail was mainly for future groups, when attempting to repeat the experimental protocol, would have a benchmark for size and viability of the co-culture model.

Supporting references (some apply to other reviewer's comments)

  1. Duan L, Ma B, Liang Y, et al. Cytokine networking of chondrocyte dedifferentiation in vitro and its implications for cell-based cartilage therapy. American Journal of Translational Research. 2015;7(2):194-208.
  2. Piñeiro-Ramil M, Sanjurjo-Rodríguez C, Rodríguez-Fernández S, et al. Generation of human immortalized chondrocytes from osteoarthritic and healthy cartilage : a new tool for cartilage pathophysiology studies. Bone and Joint Research. 2023;12(1):46-57. doi: 10.1302/2046-3758.121.Bjr-2022-0207.R1.
  3. Charlier E, Deroyer C, Ciregia F, et al. Chondrocyte dedifferentiation and osteoarthritis (OA). Biochemical Pharmacology. 2019;165:49-65. doi: https://doi.org/10.1016/j.bcp.2019.02.036.
  4. Jiang A, Xu P, Sun S, et al. Cellular alterations and crosstalk in the osteochondral joint in osteoarthritis and promising therapeutic strategies. Connective Tissue Research. 2021;62(6):709-719. doi: 10.1080/03008207.2020.1870969.
  5. Meng Y, Yin D, Qiu S, Zhang X. Abrine promotes cell proliferation and inhibits apoptosis of interleukin-1β-stimulated chondrocytes via PIM2/VEGF signalling in osteoarthritis. Phytomedicine. 2022;96:153906. doi: 10.1016/j.phymed.2021.153906.
  6. Ying H, Li Q, Zhao C. Interleukin 1β and tumor necrosis factor α promote hFOB1.19 cell viability via activating AP1. American Journal of Translational Research. 2016;8(5):2411-2418.
  7. Zhou Y, Zhou B, Pache L, et al. Metascape provides a biologist-oriented resource for the analysis of systems-level datasets. Nature Communications. 2019;10(1):1523. doi: 10.1038/s41467-019-09234-6.
  8. Garces de Los Fayos Alonso I, Liang HC, Turner SD, Lagger S, Merkel O, Kenner L. The Role of Activator Protein-1 (AP-1) Family Members in CD30-Positive Lymphomas. Cancers (Basel). 2018;10(4). doi: 10.3390/cancers10040093.
  9. O'Gorman DM, O'Carroll C, Carmody RJ. Evidence that marine-derived, multi-mineral, Aquamin inhibits the NF-κB signaling pathway in vitro. Phytother Res. 2012;26(4):630-632. doi: 10.1002/ptr.3601.
  10. Weyh C, Krüger K, Peeling P, Castell L. The Role of Minerals in the Optimal Functioning of the Immune System. Nutrients. 2022;14(3). doi: 10.3390/nu14030644.
  11. Hu T, Xu H, Wang C, Qin H, An Z. Magnesium enhances the chondrogenic differentiation of mesenchymal stem cells by inhibiting activated macrophage-induced inflammation. Scientific Reports. 2018;8(1):3406. doi: 10.1038/s41598-018-21783-2.
  12. Mastbergen SC, Lafeber FP, Bijlsma JW. Selective COX-2 inhibition prevents proinflammatory cytokine-induced cartilage damage. Rheumatology. 2002;41(7):801-808. doi: 10.1093/rheumatology/41.7.801.
  13. Ryan S, O'Gorman DM, Nolan YM. Evidence that the marine-derived multi-mineral Aquamin has anti-inflammatory effects on cortical glial-enriched cultures. Phytotherapy Research. 2011;25(5):765-767. doi: 10.1002/ptr.3309.
  14. Eriksen EF, Lech O, Nakama GY, O'Gorman DM. Disease-Modifying Adjunctive Therapy (DMAT) in Osteoarthritis-The Biological Effects of a Multi-Mineral Complex, LithoLexal(®) Joint-A Review. Clin Pract. 2021;11(4):901-913. doi: 10.3390/clinpract11040104.
  15. Li Y, Yue J, Yang C. Unraveling the role of Mg++ in osteoarthritis. Life Sciences. 2016;147:24-29. doi: https://doi.org/10.1016/j.lfs.2016.01.029.
  16. Guan P-P, Jia J-F, Wang P. Dietary magnesium ions block the effects of nutrient deficiency on inducing the formation of LC3B autophagosomes and disrupting the proteolysis of autolysosomes to degrade β-amyloid protein by activating cyclooloxygenase-2 at tyrosine 385. Journal of Functional Foods. 2021;83:104527. doi: https://doi.org/10.1016/j.jff.2021.104527.
  17. Heffernan SM, McCarthy C, Eustace S, FitzPatrick RE, Delahunt E, De Vito G. Mineral rich algae with pine bark improved pain, physical function and analgesic use in mild-knee joint osteoarthritis, compared to Glucosamine: a randomized controlled pilot trial. Complementary Therapies in Medicine. 2020;50:102349. doi: 10.1016/j.ctim.2020.102349.
  18. Bartolotti I, Roseti L, Petretta M, Grigolo B, Desando G. A Roadmap of In Vitro Models in Osteoarthritis: A Focus on Their Biological Relevance in Regenerative Medicine. Journal of Clinical Medicine. 2021;10(9). doi: 10.3390/jcm10091920.
  19. Aslam MN, Kreider JM, Paruchuri T, et al. A mineral-rich extract from the red marine algae Lithothamnion calcareum preserves bone structure and function in female mice on a Western-style diet. Calcified Tissue International. 2010;86(4):313-324. doi: 10.1007/s00223-010-9340-9.
  20. Frestedt JL, Walsh M, Kuskowski MA, Zenk JL. A natural mineral supplement provides relief from knee osteoarthritis symptoms: a randomized controlled pilot trial. Nutrition Journal. 2008;7:9. doi: 10.1186/1475-2891-7-9.
  21. Frestedt JL, Kuskowski MA, Zenk JL. A natural seaweed derived mineral supplement (Aquamin F) for knee osteoarthritis: a randomised, placebo controlled pilot study. Nutrition Journal. 2009;8:7. doi: 10.1186/1475-2891-8-7.
  22. Taylor SC, Laperriere G, Germain H. Droplet Digital PCR versus qPCR for gene expression analysis with low abundant targets: from variable nonsense to publication quality data. Scientific Reports. 2017;7(1):2409. doi: 10.1038/s41598-017-02217-x.
  23. Salgado C, Jordan O, Allémann E. Osteoarthritis In Vitro Models: Applications and Implications in Development of Intra-Articular Drug Delivery Systems. Pharmaceutics. 2021;13(1). doi: 10.3390/pharmaceutics13010060.
  24. Heffernan SM, Conway GE, McCarthy C, et al. Inflammatory markers in early knee joint osteoarthritis differ from well-matched controls and are associated with consistent, rather than intermittent knee pain. Knee. 2024;51:189-198. doi: 10.1016/j.knee.2024.09.003.
  25. Molnar V, Matišić V, Kodvanj I, et al. Cytokines and Chemokines Involved in Osteoarthritis Pathogenesis. Internal Journal of Molecular Science. 2021;22(17). doi: 10.3390/ijms22179208.

Reviewer 3 Report

Comments and Suggestions for Authors

The manuscript entitled “Red Algae extract alters expression of inflammatory pathways in a human in vitro osteoarthritis co-culture” written by Shane M. Heffernan and co-authors describes a replicable 2D co-culture model of inflammatory OA for the purpose of initial investigations of bioactive nutraceuticals. These data and the model can be used to further investigate if Lithothamnion species and other bioactive nutraceuticals can be used in tandem with pharmaceutical treatments to act synergistically. This paper generally meets the requirement of this journal. However, there are some issues in the paper that need to be addressed. Above all, I recommend this paper to be published after minor revision.

Some points

1.     All the figures in this paper should be improved. The size and style of characters should be unified. Figures with high quality should be provided.

2.     Positive control should be provided for comparison with that of red Algae extract.

3.     In figures 4G, 4H and 5G, the error is large, which should be discussed in this manuscript.

4.     Other minor points:

Line 14, species extract However--- species extract. However

Lines 25-27, please check and revise this sentence.

Lines 27, Lithothamnion species--- Lithothamnion species. “species” uses normal style. Please check the whole manuscript.

Author Response

Comments and Suggestions for Authors

The manuscript entitled “Red Algae extract alters expression of inflammatory pathways in a human in vitro osteoarthritis co-culture” written by Shane M. Heffernan and co-authors describes a replicable 2D co-culture model of inflammatory OA for the purpose of initial investigations of bioactive nutraceuticals. These data and the model can be used to further investigate if Lithothamnion species and other bioactive nutraceuticals can be used in tandem with pharmaceutical treatments to act synergistically. This paper generally meets the requirement of this journal. However, there are some issues in the paper that need to be addressed. Above all, I recommend this paper to be published after minor revision.

Some points

  1. All the figures in this paper should be improved. The size and style of characters should be unified. Figures with high quality should be provided.

Thank you for your comment. Complete.

  1. Positive control should be provided for comparison with that of red Algae extract.

The Authors would like to thank the Reviewer for this comment. The extract used throughout the study was derived from a commercially available proprietary extraction method (as described in the manuscript). The material has a naturally occurring organic multimolecular complex with a unique porous microstructure that is maintained through the processing technique (reference 60 and 61 in the manuscript) and as such there is no potential material/chemical that could act as a positive control. However, as much as was reasonable possible, all care was taken to control the experimental conditions to ensure accurate data and interpretation, as well as insuring appropriate positive and negative assay controls were completed where appropriate.

  1. In figures 4G, 4H and 5G, the error is large, which should be discussed in this manuscript.

Thank you for your comment. The Authors would politely disagree that the error is large in Figure 4H. However, the large variability in Figure 4G could be due to using the unstimulated co-culture model and the differences in IL-8 production in the different cell types25. The large error bars in 5G could be due to the utilisation of the co-culture and multiple cell types having TNFS10 expression measured. This is also at day 7 where the differences in gene expression may be starting to be more apparent at this time point for this gene, when compared to day 1. However, given the reviewers comment additional text have been added to the discussion highlighting the larger error of some outcome variables. This now reads;

“Some larger error were present in individual outcome measures; however these were not completely unexpected given the use of multiple cell lines, inflammatory stimulus and treatment with an organic material. Nonetheless, additional caution is required when interpreting these data.”

  1. Other minor points:

Line 14, species extract However--- species extract. However

Thank you for your comment. Complete.

Lines 25-27, please check and revise this sentence.

Thank you for your comment. The text has now been revised and now reads.

“These data present a novel and replicable co-culture model of inflammatory OA that can be used to investigate bioactive nutraceuticals. For the first time, this model demonstrated a reduction of C1qTNF2 that can be mitigated with Red Algae Lithothamnion species extract.”

Lines 27, Lithothamnion species--- Lithothamnion species. “species” uses normal style. Please check the whole manuscript.

Thank you for your comment. This has now been corrected throughout the manuscript.

Supporting references (some apply to other reviewer's comments)

  1. Duan L, Ma B, Liang Y, et al. Cytokine networking of chondrocyte dedifferentiation in vitro and its implications for cell-based cartilage therapy. American Journal of Translational Research. 2015;7(2):194-208.
  2. Piñeiro-Ramil M, Sanjurjo-Rodríguez C, Rodríguez-Fernández S, et al. Generation of human immortalized chondrocytes from osteoarthritic and healthy cartilage : a new tool for cartilage pathophysiology studies. Bone and Joint Research. 2023;12(1):46-57. doi: 10.1302/2046-3758.121.Bjr-2022-0207.R1.
  3. Charlier E, Deroyer C, Ciregia F, et al. Chondrocyte dedifferentiation and osteoarthritis (OA). Biochemical Pharmacology. 2019;165:49-65. doi: https://doi.org/10.1016/j.bcp.2019.02.036.
  4. Jiang A, Xu P, Sun S, et al. Cellular alterations and crosstalk in the osteochondral joint in osteoarthritis and promising therapeutic strategies. Connective Tissue Research. 2021;62(6):709-719. doi: 10.1080/03008207.2020.1870969.
  5. Meng Y, Yin D, Qiu S, Zhang X. Abrine promotes cell proliferation and inhibits apoptosis of interleukin-1β-stimulated chondrocytes via PIM2/VEGF signalling in osteoarthritis. Phytomedicine. 2022;96:153906. doi: 10.1016/j.phymed.2021.153906.
  6. Ying H, Li Q, Zhao C. Interleukin 1β and tumor necrosis factor α promote hFOB1.19 cell viability via activating AP1. American Journal of Translational Research. 2016;8(5):2411-2418.
  7. Zhou Y, Zhou B, Pache L, et al. Metascape provides a biologist-oriented resource for the analysis of systems-level datasets. Nature Communications. 2019;10(1):1523. doi: 10.1038/s41467-019-09234-6.
  8. Garces de Los Fayos Alonso I, Liang HC, Turner SD, Lagger S, Merkel O, Kenner L. The Role of Activator Protein-1 (AP-1) Family Members in CD30-Positive Lymphomas. Cancers (Basel). 2018;10(4). doi: 10.3390/cancers10040093.
  9. O'Gorman DM, O'Carroll C, Carmody RJ. Evidence that marine-derived, multi-mineral, Aquamin inhibits the NF-κB signaling pathway in vitro. Phytother Res. 2012;26(4):630-632. doi: 10.1002/ptr.3601.
  10. Weyh C, Krüger K, Peeling P, Castell L. The Role of Minerals in the Optimal Functioning of the Immune System. Nutrients. 2022;14(3). doi: 10.3390/nu14030644.
  11. Hu T, Xu H, Wang C, Qin H, An Z. Magnesium enhances the chondrogenic differentiation of mesenchymal stem cells by inhibiting activated macrophage-induced inflammation. Scientific Reports. 2018;8(1):3406. doi: 10.1038/s41598-018-21783-2.
  12. Mastbergen SC, Lafeber FP, Bijlsma JW. Selective COX-2 inhibition prevents proinflammatory cytokine-induced cartilage damage. Rheumatology. 2002;41(7):801-808. doi: 10.1093/rheumatology/41.7.801.
  13. Ryan S, O'Gorman DM, Nolan YM. Evidence that the marine-derived multi-mineral Aquamin has anti-inflammatory effects on cortical glial-enriched cultures. Phytotherapy Research. 2011;25(5):765-767. doi: 10.1002/ptr.3309.
  14. Eriksen EF, Lech O, Nakama GY, O'Gorman DM. Disease-Modifying Adjunctive Therapy (DMAT) in Osteoarthritis-The Biological Effects of a Multi-Mineral Complex, LithoLexal(®) Joint-A Review. Clin Pract. 2021;11(4):901-913. doi: 10.3390/clinpract11040104.
  15. Li Y, Yue J, Yang C. Unraveling the role of Mg++ in osteoarthritis. Life Sciences. 2016;147:24-29. doi: https://doi.org/10.1016/j.lfs.2016.01.029.
  16. Guan P-P, Jia J-F, Wang P. Dietary magnesium ions block the effects of nutrient deficiency on inducing the formation of LC3B autophagosomes and disrupting the proteolysis of autolysosomes to degrade β-amyloid protein by activating cyclooloxygenase-2 at tyrosine 385. Journal of Functional Foods. 2021;83:104527. doi: https://doi.org/10.1016/j.jff.2021.104527.
  17. Heffernan SM, McCarthy C, Eustace S, FitzPatrick RE, Delahunt E, De Vito G. Mineral rich algae with pine bark improved pain, physical function and analgesic use in mild-knee joint osteoarthritis, compared to Glucosamine: a randomized controlled pilot trial. Complementary Therapies in Medicine. 2020;50:102349. doi: 10.1016/j.ctim.2020.102349.
  18. Bartolotti I, Roseti L, Petretta M, Grigolo B, Desando G. A Roadmap of In Vitro Models in Osteoarthritis: A Focus on Their Biological Relevance in Regenerative Medicine. Journal of Clinical Medicine. 2021;10(9). doi: 10.3390/jcm10091920.
  19. Aslam MN, Kreider JM, Paruchuri T, et al. A mineral-rich extract from the red marine algae Lithothamnion calcareum preserves bone structure and function in female mice on a Western-style diet. Calcified Tissue International. 2010;86(4):313-324. doi: 10.1007/s00223-010-9340-9.
  20. Frestedt JL, Walsh M, Kuskowski MA, Zenk JL. A natural mineral supplement provides relief from knee osteoarthritis symptoms: a randomized controlled pilot trial. Nutrition Journal. 2008;7:9. doi: 10.1186/1475-2891-7-9.
  21. Frestedt JL, Kuskowski MA, Zenk JL. A natural seaweed derived mineral supplement (Aquamin F) for knee osteoarthritis: a randomised, placebo controlled pilot study. Nutrition Journal. 2009;8:7. doi: 10.1186/1475-2891-8-7.
  22. Taylor SC, Laperriere G, Germain H. Droplet Digital PCR versus qPCR for gene expression analysis with low abundant targets: from variable nonsense to publication quality data. Scientific Reports. 2017;7(1):2409. doi: 10.1038/s41598-017-02217-x.
  23. Salgado C, Jordan O, Allémann E. Osteoarthritis In Vitro Models: Applications and Implications in Development of Intra-Articular Drug Delivery Systems. Pharmaceutics. 2021;13(1). doi: 10.3390/pharmaceutics13010060.
  24. Heffernan SM, Conway GE, McCarthy C, et al. Inflammatory markers in early knee joint osteoarthritis differ from well-matched controls and are associated with consistent, rather than intermittent knee pain. Knee. 2024;51:189-198. doi: 10.1016/j.knee.2024.09.003.
  25. Molnar V, Matišić V, Kodvanj I, et al. Cytokines and Chemokines Involved in Osteoarthritis Pathogenesis. Internal Journal of Molecular Science. 2021;22(17). doi: 10.3390/ijms22179208.